# How Did Bhikṣuṇī Meet Indian Astrology? Viewing the Buddhist Narration and Logic from the Story of the Mātaṅga Girl

**Liqun Zhou**

Department of Sanskrit and Pali, School of Asian Studies, Beijing Foreign Studies University, Beijing 100089, China; liqun21cn@bfsu.edu.cn

**Abstract:** The story of *Śārdūlakarṇāvadāna* consists of stories of the present life and past life. The former is about a girl from the low-caste *Mātaṅga* tribe who pursues Ananda, a disciple of the Buddha, but her pursuit ends in vain, and she eventually converts to Buddhism. The latter is about a low-caste king demonstrating his knowledge of the Vedas and astrology in a bid to marry the daughter of a great Brahmin for his son. The story could be seen in various Buddhist texts, such as the *Divyavadāna* from Nepal and the *Mātaṅga Sutra* in China. This paper studies the narration and logic of *Śārdūlakarṇāvadāna* stories, and it makes conclusions on the commonalities in the compilation of Buddhist narratives by analyzing the beginning and end of the story, as well as its narrator, narratee, and the four conflicts (i.e., the caste barriers, the violation of precepts, the use of incantations, and the use of expertise in seeking marriage).

**Keywords:** *Mātaṅga*; narrator; narratee; conflicts in storyline

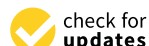

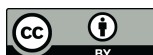

## 1. Introduction

The modern Indian writer Rabindranath Tagore's dance drama, *The Chandalika*, Shang Xiaoyun's Peking Opera *Modengnü* (摩登女), and Zhang Daqian's copy of the ancient fresco *Modengnü* (摩登女) all coincidentally focus on the female protagonist the Mātaṅga girl from the sutra *Śārdūlakarṇāvadāna, The story of Tiger's Ear* in English (Zhou 2020, pp. 3–4). The modern Chinese word "mó dēng (摩登)", a transliteration of the English word "modern", is indeed borrowed from the ancient Chinese translation of the sutras with its cultural connotations of Indian women (Zhang 2007, p. 31). Both the figure and the vocabulary are associated with the *Śārdūlakarṇāvadāna* (in short, the ZKA), which was translated into Chinese around the fourth century AD.

In the second volume of *A History of Indian Literature*, published in 1913, Winternitz, a leading German expert on Indian literature, refers specifically to the ZKA story as one of the most impressive stories in the *Divyāvadāna*. Throughout many years of research, I have been amazed at the delicate combination of Buddhist literature and astronomy in this story. It seems that no scholar has yet responded squarely to the specific reasons why the ZKA story is so interesting. The following elements of the story are all important in shaping the text: the young disciple of the Buddha, Ananda, being courted by a young girl; the attempt of a young girl from a lower caste to love beyond caste and religion; the correspondence between the four main characters in the past-life story and the present-life story; the father in the past-life story showing his great knowledge of all ancient India in order to help his son in his courtship; and the knowledge of astrological divination, namely, the astronomical knowledge of ancient India, narrated by a low-caste king. This paper attempts to discuss in detail the narration and logic of the *Śārdūlakarṇāvadāna*, with a view to examining the commonalities and individuality of the text.

The ZKA contains two metaphorical parts: one of the past life and the other of the present life. The present part[1] focuses on the story of Bodhisattva's disciple, Ananda, who

was pursued by the Bodhisattva during the Buddha's lifetime and was converted into the monastic community. The second is about Bodhisattva's past life. In the previous life as King *Triśaṅku* of the *Mātaṅga* tribe, the Buddha wished for the great Brahmin named *Puṣkarasāriṇ* (the previous life of the *Mātaṅga* girl's mother) to marry his daughter, Prakṛti (the previous life of the *Mātaṅga* girl), for his son, *Śārdūlakarṇa* (the previous life of Ananda), but this was objected to. The Buddha told King *Puṣkarasāriṇ* and his subjects that this woman had been married to Ananda in all her previous five hundred lives, which is why she fell in love with him in this life and pestered him so much. Finally, she converted to the Buddhist order and became a member of the monastic community.

There are many versions of the story containing the present life, e.g., the Chinese translation, which includes. Ch 1 of the ZKA, i.e., *Modengjia jing* (supposed to be translated by Zhulvyan and Zhiqian) 摩登伽經 （題為竺律炎+ 支谦譯）, chp. 2 of the ZKA, i.e., *Shetoujian taizi ershibaxiu jing* (translated by Dharmarakṣa Zhufahu) 舍頭諫太子二十八宿譯 (竺法護譯), *Foshuo modengnü jing* (supposed to be translated by An Shigao) 佛说摩邓女经 （題為安世高譯）, and *Foshuo modengnü jiexing zhong liushi jing* (anonymous) 佛說摩登女 解形中六事經 （失譯）; the Nepalese Sanskrit critical edition in the 17th century *Śārdūlakar ṇāvadāna*; and the Tibetan translation in the ninth-century *sTag rNa'i rTogs pa brJod pa*. The story is widely circulated in various Buddhist texts, such as the Central Asian Sanskrit text *Merv avadāna*, the Sanskrit version *Divyāvadāna* from Nepal, and *Binaiye* (鼻奈耶), all of which fully exemplify the characteristics of the *Divine Stories*.

The main content of the Oxford Sanskrit text of the ZKA is the story of the past life of Ananda and the Mātaṅga girl. The *Triśaṅku*, the King of *Mātaṅga*, uses his learning and knowledge to debate with the great Brahmin Puṣkarasāriṇ about the marriage of his daughter and Prince Śārdūlakarṇa (Tiger's Ear). The debate centered on whether Caṇḍāla, as a bastard caste outside the four castes, was qualified to ask for a Brahmin daughter. Brahmin families were prominent in ancient India and had the privilege of mastering the transmission of certain knowledge for generations, such as those who recited the Yajurveda for generations and those who had knowledge of astrology for generations. The topic that Triśaṅku and Puṣkarasāriṇ discussed contains not only the origins of the Vedas but also the knowledge of the 28 lunar mansions. There are a few parallel texts of ZKA containing stories from the past lives: the Chinese translations of the ZKA, i.e., *Shetoujian taizi ershibaxiu jing* (translated by Dharmarakṣa Zhufahu) 舍頭諫太子二十八宿經 （竺法護譯）; the seventeenth-century Nepali Sanskrit text of the ZKA; and the ninth-century Tibetan translation of the ZKA.

## 2. The Narration and Logic of *Śārdūlakarṇāvadāna*
### 2.1. The Opening and Ending Phrases

The narrative style of Buddhist texts has a certain paradigm. It is generally accepted that Buddhist scriptures begin with the words "Thus have I heard", which is the original text of the Buddha's words as heard by Ananda, i.e., "Buddhavacana". The Sanskrit text of the ZKA begins with "oṃ, namo ratnatrāyāya. evaṃ mayā śrutam". "Om salute the Three Treasures, thus have I heard". The other versions of the *ZKA* and the Tibetan Sanskrit text have only the phrase "Thus have I heard", and none of them has the phrase "Om salutes the Three Treasures". The *Modejia jing* (摩登伽經) begins with the phrase "As have I heard (如是我聞)", and the *Shetoujian taizi ershibaxiu jing* (舍頭諫太子二十八宿經) begins with the phrase "Heard as this (聞如是)", but, although the Chinese translations are slightly different, they also do not contain the phrase "Salutations to the Three Treasures". The Sanskrit text on which the ZKA is based is mainly the 1886 Sanskrit text written in the Bengal district and the 1837 Sanskrit text given by Hodson to the Asiatic Society in Paris, and it is likely that the phrase "saluting the Three Treasures" was added by a nineteenth-century scribe. In both Chinese translations, the phrases "heard as this (聞如是)" and "Thus have I heard (如是我聞)" are translations of the Sanskrit phrase "evaṃ mayā śrutam". It is generally believed that "heard as this (聞如是)" appears in an earlier translation of the sutra, while "Thus have I heard (如是我聞)" appears in a later translation, used by translator Yi

Jing (義淨) from the Tang Dynasty and translator Shi Hu (施护) from the Song (宋) Dynasty, gradually becoming a standard in the translation of Buddhist texts.

The Sanskrit critical edition of the ZKA from Nepal ends the same way as the edition of the ZKA in *Divyāvadāna*. "*idam avocad bhagavān/āttamanasas te bhikṣavo bhagavato bhāṣitamabhyanandan//Iti śrīdivyāvadāne śārdūlakarṇāvadānam//*". The English translation is as follows: "After His Holiness had said this, the bhikṣus rejoiced at His words. The above is the śārdūlakarṇāvadāna in auspicious Divyāvadāna". The end of Ch 2 of the ZKA reads, "The Buddha said thus. And the king, Prasenajit, was overwhelmed with joy and enthusiasm, and the elder Brahmacharyas and the bhikṣus saluted the Buddha. the *Shetoujian taizi ershibaxiu jing* (舍頭諫太子二十八宿經)" (Ch 2 of the ZKA, vol. 21, 419c25–c27). The Sutra of the Bodhisattva ends with the following words: "The Buddha said this sutra. The king of Prasenajit and the four tribes followed it with joy. *The* second volume of *the* Modengjia jing (摩登伽經)" (Ch 1 of ZKA, vol. 21, 410b12–b13). The Sanskrit scribe clearly knew that the Sanskrit text of the *ZKA* came from the *Divyāvadāna*, whereas the compilers of Ch 1 and Ch 2 of the ZKA may not have seen the collection of the *Divyāvadāna* and did not mention that the story came from the *Divyāvadāna*. In the Sanskrit text, the concluding parts do not mention King Prasenajit, whereas, in both the above two Chinese translations, he is explicitly mentioned. At the end of Ch 2 of the ZKA, more details are added, and the people who were delighted by the Buddha's words were not only the *bhikṣu*s and King *Prasenajit*, but also other "elderBrahmacharyas", i.e., brahmins and householders; after hearing the story of the parable, in addition to being delighted, they also "saluted the Buddha" by paying homage to him. They were delighted to hear the story of the parable, and they also "saluted the Buddha".

The *Avadāna* is one of the nine or twelve sutras, pronounced *Abotuona* (阿波陀那), and it is formally identical to the Jātaka. Broadly speaking, the *Avadāna* includes Buddhist literature, Buddhist praise, and karmic stories. The Sanskrit text is represented by the *Avadānaśataka*, the *Divyāvadāna*, and the *Jātakamālā*, and its Chinese translation includes *The Sutra of Virtues and Fools* (賢愚經), *The Sutra of the Six Degrees* (六度集經), *The Sutra of the Hundred Metaphors* (百喻經), *The Sūtra of Collected Hundred Occasions* (撰集百缘經*Avadānaśātaka*), *The Discourse on the Bodhisattva's Origin* (菩薩本生鬘), *The Miscellaneous Treasure Sutra* (大莊嚴經論), *The Discourse on the Great Sutra* (大宝藏經), and *The Bodhisattva Ben Yuan Sutra* (菩薩本缘經). Venerable Yin Shun (印順) and Ding Min (丁敏) (Yinshun 印順 1999, p. 460; Ding 1996, p. 71) classified the *avadāna* stories into categories, namely, *avadāna-itivṛttaka*, *avadāna-jātaka*, *vyākaraṇa*, and *avadāna-vyākaraṇa*, presenting *avadāna*. According to their classification, the ZKA is the type that connects the stories of this life with the ones of the past, i.e., *avadāna-itivṛttaka* (譬喻本事). Since the past-life story features the incarnations of the Buddha and his disciple Ananda, the ZKA is said in a narrow sense to be an *avadāna-jātaka* (譬喻本生). In a broader sense, this life is a type of metaphorical skill, and as the word "metaphor" appears in the title of the ZKA, it is considered a type of metaphorical skill according to Ding Min's classification. According to Fan Jingjing's research, The Sutra of the Heavenly Parables is, unlike the others, an anthology of parables scattered throughout the sutras and rituals. In his doctoral thesis, Tsutomu Matsumura divides the development of metaphorical literature into four stages: first, it became entangled with the sutra and the tales of the present life; then, it became a separate branch of the twelve divisions, but of such a variety that it was not yet linked to the concept of the same; and next, stories were selected from the canonical stories and other texts (e.g., the Great Sutra treatises) to form collections of stories, such as the Celestial Metaphor Sutra, to which the metaphorical stories of Gilgit belong. It is from *The Sūtra of Collected Hundred Occasions* (撰集百缘經) onwards that the metaphorical literature takes a fully independent path of development.[2]

### 2.2. Narrator

In cases where the narrator is also a literary character, they may play a role of greater or lesser importance in the events that they are telling. They may be the main character, an important character, a minor character, or even a mere bystander; sometimes, they may be

a character only in their part of the narrative and be absent in others; sometimes, although they do not play a role in the events that they tell, they may also be a character in the events told by other narrators (Shanruzod in *The Thousand and One Nights*). There are narrators who are also male protagonists, as in the case of the male protagonists in literary works such as *The Consciousness of Zeno*, *Great Expectations*, and *Kiss me deadly* (Prince 1982, p. 15).

In various Buddhist artistic subjects, the handsome young man standing at the right hand of the Buddha is His Holiness Ananda (*Ānanda*). Ananda followed and served the Buddha for a long time, and Buddhist texts record that he fanned the Buddha to bring refreshment and brought him cool water to quench his thirst, making him Buddha's closest disciple. As he accompanied the Buddha for a long time, naturally, he heard the Dharma the most. After the Buddha's nirvana, the monks recited the teachings that they had heard at the first gathering so that they would not die out. Many sutras begin with Ananda asking the Buddha a question that brings up the rest of the story. Ananda remembered and recited the most sutras, so everyone called him "the first to hear much (多聞第一)". Young, handsome, erudite, knowledgeable, and single, Ananda seemed to be the ideal male figure for young girls in ancient India, where literacy rates were low.

*2.3. Narratee*

If there is at least one narrator in any narrative, then there is also at least one narratee, who may or may not be explicitly referred to as "you". In many narratives where the narratee is not referred to as "you", the "you" may be removed without trace, leaving only the narrative itself (Prince 1982, p. 17).

Apart from the *dārṣṭāntika*s who preach *avadāna* stories at great festivals, these stories are also told in the homes of lay people during pujas. Andy Rotman summarizes several situations in which stories are told in *Divyāvadāna*: Firstly, stories are told when answering questions for the monastic community and telling stories of karma. Secondly, stories are told when preaching the *Dharma* to the lay congregation who come to hear it. Thirdly, the Dharma is told after receiving offerings in the homes of lay people, which Rotman speculates is the *avadāna* story (Rotman 2008). In ancient India, the distinction between a *dārṣṭāntika* and a chanting teacher is very clear. The former used various illustrations, parables, stories, and, in later times, stories of karma in particular in order to spread doctrine and compile a group of classics. The latter, however, were mainly responsible for the consecration and glorification of the Buddha. In ancient China, there was also a separation between the "chanting of the Buddha's name" and the "preaching of the Buddha's teachings in a miscellaneous sequence of causes and effects, and the quoting of *avadāna* story in the background". However, since the brilliance of Huiyuan (慧遠), the singing teacher has been required to speak about the "three lives of karma ". It has since become customary for the chanting teacher to be the teacher of *avadāna* stories (Fan 2020, pp. 126, 136).

The Sanskrit and Chinese versions of the ZKA, which tells the story of the cycle of karma in two lives, may have been the versions used by *dārṣṭāntika*s in India and in China. The story of the past lives in the ZKA was told to King *Prasenajit* and many of his ministers, who listened and followed it with joy. In fact, the whole story of the ZKA and, indeed, the entire text were created for ordinary people with an interest in Buddhism. In the ZKA story, there are two or more narrators, and it is one of them to whom the whole story is ultimately addressed. In contrast to the other narratees, the ZKA addresses King *Prasenajit*, who was in a position of power.

## 3. Conflicts in the Main Plot of the Story

The ZKA has been described as the most interesting story by Winternitz, the author of *A History of Indian Literature*, probably because the conflicts in the storyline are more obvious and more frequent, thus creating several interesting story conflicts, such as the caste barrier, the violation of precepts, the use of incantations, and the use of expertise in seeking marriage, which produces several points of interest.

### 3.1. The Conflict of Caste Barriers

The heroine of the story is called the *Mātaṅga* girl, which translates as *modengjia* (摩登伽), or *modengnü* (摩登女), etc. India has a complex system of castes and tribes, and *Mātaṅga* is the name of a tribe that belongs to the caste of *Caṇḍala* (sometimes spelled Chandala). The *Caṇḍala* caste is known as the "lowest of the low", well below the Brahmins (*Brāhmaṇa*), Kshatriyas (*Kṣatriya*), Vaishyas (*Vaiśya*), and Shudra (*Śudra*) castes. Usually, the offspring of a union between a high-caste Brahmin woman and a Shudra man were called *Caṇḍala*[3], an outcaste (Varṇasaṃkara). This is because ancient Hindu Dharma literature, such as *Manusmṛti*, states that the marriage of a woman of a lower caste to a caste higher than her own is considered a civil marriage and that marriage to a man of a lower caste is considered a reverse marriage. A man of the *Caṇḍala* caste, regardless of which caste he intermarries with, gives birth to a woman of the *Caṇḍala* caste, meaning that once a Caṇḍala is born from a reverse marriage, the offspring will always be *Caṇḍala* as well.

In ancient India, occupation, residence, possessions, food, and dress were closely linked to caste and tribe, and the *Caṇḍala* caste, the "lowest of the low", lived in the worst conditions. The *Manusmṛti* states that "the dwelling place of *Caṇḍala*s must be outside the village, they must be treated as mendicants, and their possessions must be dogs and donkeys. They must wear the clothes of dead, they must eat food from broken plates, their ornaments must be made of iron, and they must wander forever. Those who practice the law must not have a desire to associate with them; their affairs must be done within them; they must intermarry with people of their own kind. By day they must go out to do their business after they have been marked by the king's command; they must carry the dead bodies of those who have no relatives; these are the usual conditions. They must always follow the king's order to execute the guilty according to the rules; they must take away the clothes, bedclothes and ornaments of the executed. The untouchable, the unknown, the impure, eve' if outwardly Aryan, but not actually Aryan, must be determined by their own conduct. Vulgarity, rudeness, cruelty and a habit of not observing one's duty are the characteristics of the impure-blooded people of this world" (Jiang 2007, p. 212). The caste system was like a great net that strictly limited the *Caṇḍala*s' food, clothing, and shelter so that they lived their lives as humble as ants.

The *Mātaṅga* tribe, so the story told, was responsible for repairing people's carts on the roads. When it was not harvesting time, to earn enough money, they would put sharp stones on the road and deliberately break the wheels of vehicles, and then they would repair the vehicles for money[4]. This kind of business is common in poor and inaccessible places. The *Mātaṅga* girl of the *Caṇḍala* caste, born into a family of Brahmins and Shudras married against their will, may live outside the village, taking off her shoes, wearing the clothes of the dead, carrying iron ornaments, and eating food from broken plates when she enters the village. She wishes to invite other girls to her house and entertain them with good food and drink; girls of a higher caste than her are reluctant to go. Whenever she steps outside her house, she is discriminated against in many ways, large and small. The only light in life is that the mother of the modem girl is a Brahmin woman with knowledge and the ability to cast mantras and to protect her and give her comfort if others bully her.

The meeting between Ananda and the *Mātaṅga* girl in this life was purely fortuitous. According to the precepts of the Buddha's time, monks in India did not cook their own food but went out in the morning to beg for food with a bowl. On his way back to his monastery, Ananda felt thirsty, so he went to a large pool and asked a girl who was fetching water for a drink. The girl who was fetching water was none other than the *Mātaṅga* girl, who at first refused Ananda's request, saying that she was the *Caṇḍala* caste and was not fit to give water to a passenger. Ananda said that he was a Buddhist monk, that there was no inferiority or superiority in his heart, and that all men were equal. He pleaded with the girl to give him water to drink as soon as possible, after which he had to continue his journey. The *Mātaṅga* girl could not resist Ananda, so she fetched water for him from a pitcher, and when he had finished drinking, Ananda left rapidly. Ananda returned to the abbey to recite the sutra and meditate, but the *Mātaṅga* girl fell in love with Ananda from this

brief encounter. She liked Ananda's looks, voice, words, and even the way he raised his hands. This would have been unthinkable in another time and place, but it would have made sense in India, where the caste system was so rigid. As mentioned earlier, people from the lower castes are discriminated against everywhere; they have to live outside the village, they have to take off their shoes to enter the village, and few shops will sell them garlands, milk, or other daily necessities. As the lowest of the low castes, the *Caṇḍala*s were never given the opportunity to eat or drink with the higher castes, let alone provide them with drinking water. In order to keep their holiness untainted by the lower castes, the higher-caste people would not accept food and water from the lower castes in any case. So, Ananda's act of drinking from the *Mātaṅga* girl's water jar broke the barrier of caste in her mind and made her feel recognized and accepted, feeling that this handsome monk was like a heavenly god. It is not surprising that the *Mātaṅga* girl was attracted to Ananda and wanted to be with him.

### 3.2. Conflicts against Buddhist Precepts

As a monk, Ananda was not allowed to enter secular life, such as marrying and having children. Embarrassed and frightened by the approval and courtship of the *Mātaṅga* girl, Ananda kept fleeing. But the *Mātaṅga* girl followed Ananda into the city and begged for food, walking as Ananda walked and standing as Ananda stood. This is a rather unorthodox situation for a Buddhist monk or a secular family. When Ananda ignored the *Mātaṅga* girl, the girl's mother used a mantra to catch Ananda and make him walk into their home, trying to let them get married in order to keep her daughter alive. In the nick of time, the Buddha used his magical powers and learned that Ananda was confined; he used a mantra to break the Brahmin woman's mantra, and Ananda was able to return to the monastery. The *Mātaṅga* girl had no other choice but to stay at the door of the monastery. Such a thing would still be inappropriate for a monk. Buddha himself spoke to the *Mātaṅga* girl and her family and told her that she could only be with Ananda if she became a *bhikṣuṇī* and joined the Buddhist Sangha. In fact, even after becoming a *bhikṣuṇī*, the *Mātaṅga* girl could not be with Ananda in the same way as a couple in secular life. Knowing all this, she left her parents and became a *bhikṣuṇī* in order to be closer to her beloved.

Generally, a *bhikṣu* takes hundreds of precepts, some of which forbid one to "change one's mind through lust". A *bhikṣu* is forbidden from having physical contact with a woman, speaking intimately with a woman, fornicating with a woman, preaching too much for a woman, sitting alone with a woman, staying in the same house with a woman, walking on the same path as a woman, being in close distance to a *bhikṣuṇī*, to walk together, or to travel in a boat together. It is clear from the scope of these precepts that the modem woman trailing Ananda, wishing him to be her husband, and guarding the door of the monastic residence had already seriously violated the precepts. The Buddha, as the leader of the monastic community, could not have allowed her to continue. It was a more feasible solution to involve the *Mātaṅga* girl in the monastic community, to ordain her, to speak to her, and to bind her by the precepts of the monastic community. The Buddha had the *Mātaṅga* girl shave and dress in monk's clothing, with the dharma name *Prakṛti*, after which he gave her a discourse on the Four Noble Truths of Suffering, Concentration, and Destruction. As a *bhikṣuṇī*, the Bodhisattva was enlightened and attained the Four Noble Truths and the fruit of Arahatship. Previously, there had been no women in the Buddhist monastic community, and the inclusion of women in the community of *bhikṣuṇī*s shocked King *Prasenajit* and his subjects. The Buddha had to explain the matter, so he told the story of how the *Mātaṅga* girl and Ananda had been husband and wife for five hundred lifetimes, hoping to gain an understanding of the monastic community and the secular crowd.

### 3.3. The Conflict over the Use of Mantras to Capture Ananda

The *Mātaṅga* girl was the daughter of a Brahmin woman and was, therefore, intelligent and clever. Knowing that she could not easily marry Ananda, she went home and asked her mother, who was skilled in mantras, for help. Naturally, her mother refused to help at

first, fearing that doing so would bring about the destruction of her family. The *Mātaṅga* girl pleaded bitterly and repeatedly expressed that she could not live without Ananda. Despite her displeasure, her mother had no choice but to relent and grant her request. The mother painted the floor of her house with cow dung and covered it with white thatch, and she made a large fire in the middle of the house. She took one hundred and eight flowers of the magical curse and chanted a mantra as she circled the fire, throwing one flower into the fire after each recitation. Her mantra was

> "Pure and stainless saffron and jasmine! Where you are bound, there is lightning. The god sends forth rain, lightning and thunder as he wishes. To astonish the great king as well as gods, men and gandharvas—O gods of planets with fire and gods of planets without fire!—and so that Ānanda shall return, meet with, approach and embrace Prakṛti, I perform this ritual."

> "*amale vimale kuṅkume sumane/yena baddhāsi vidyut/icchayā devo varṣati vidyotati garjati/vismayaṃ mahārājasya samabhivardhayituṃ devebhyo manuṣyebhyo gandharvebhyaḥ śikhigrahā devā viśikhigrahā devā ānandasyāgamanāya saṃgamanāya kramaṇāya grāṇāya juhomi svāhā//*".[5]

After the mantra was cast on Ananda, his mind was confused, and he went into ecstasy and unconsciously came towards the home of the *Mātaṅga* girl. When Ananda arrived, he saw her making her bed and suddenly had an awakening, yet he was still unable to control his body and wept in pain. He hoped that the Great Compassionate Buddha would get him out of his suffering. Seeing Ananda's plight with his celestial eyes, the Buddha recited the six mantras:

> "Fortitude, stalwartness, good conduct and safety to all living beings! Let this clear, pure, calm mind-stream bring to all a fearlessness. In which all calamities, dangers and disturbances are quelled, and to which gods, yogins and all adepts pay homage—by the truth of this speech, may the monk *Ānanda* be safe."

> "*sthitir acyutiḥ sunītiḥ/svasti sarvaprāṇibhyaḥ//saraḥ prasannaṃ nirdeśaṃ praśāntaṃ sarvato 'bhayam/ītayo yatra śāmyanti bhayāni calitāni ca//tadvai devā namasyanti sarvasiddhāśca yoginaḥ/etena satyavākyena svastyānandāsya bhikṣave//*",[6]

Then, he used his divine power to help Ananda escape from the grip of the Mātaṅga woman and her daughter and return home.

According to ancient texts of India, people believe in the power of the body, speech, and mind. In Hindu mythology and literature, episodes in which vows and mantras become a reality and cause great hurt to mundane people and even celestial gods are commonly used. In the Ramayana, for example, Rama does not believe in *Sītā*'s chastity, and the many seers make *Sītā* swear to Rama to prove her innocence. "Looking at all the people, *Sītā*, dressed in a yellow robe; his eyes downcast and his head bowed, folded his hands and said: 'If I have never wanted any man but Rama; then ask the goddess of the earth to show a gap for me to enter.' *Sītā* thus vowed, and an unexpected miracle occurred; the lioness of the supreme heavens sprang up before him in the earth. And the great dragon of infinite strength, with his head, brought up the throne; and this throne came from heaven, and all the treasures of heaven were made. Then the goddess, the Mother of the Earth, put her arms together and embraced *Sītā*; saluting and welcoming her, she placed her on the throne. And Siddhartha sat on the throne, and all at once she fell into the earth; and scattering to *Sītā*, a continual rain of flowers fell into the blue sky. All at once the gods of heaven shouted, "Goodness!" And the cries did not stop, "Goodness! Goodness!" They cried out, "How virtuous *Sītā* is. When they saw *Sītā* enter into the earth, the gods were overjoyed; and as they spoke thus, they all returned one by one to their heavenly palaces" (Ji 1984). After *Sītā* had made her vow, Mother Earth welcomed her into the Earth, and the vow was fulfilled, and Rama could not see her again. Buddhism also believes in the power of the body, speech, and mind, and there is a tradition of using mantras in Hindu Tantra, Tibetan Tantric Buddhism, and Japanese Shingon Buddhism. For the layman, the

mantras in the ZKA have a mystical quality and have powers beyond nature. The mantras in the story are worthy of being phonetically translated into different languages as they were originally written. Buddhists or secular people, reciting them according to the text, may have magical powers.

*3.4. The Story Conflict of Seeking Marriage with Specific Knowledge*

In the past-life story, the conflict focuses on how a low-caste king can marry a daughter of a great Brahmin for his son. The story of the *Mātaṅga* girl profoundly demonstrates the deep-rootedness of the caste system in India. As a king, *Triśaṅku* had power, influence, and property. His son, *Śārdūlakarṇa* (Tiger-ear), was a man of great wisdom, good looks, and fine conduct, and of a compassionate and gentle disposition, possessing all the rare virtues. When the great Brahmin *Puṣkarasārin* heard that the *Caṇḍala* King wanted to marry his own talented and virtuous daughter for his son, he felt that the marriage proposal was an insult to the highest caste by the lower caste and became angry and rebuked him for telling *Triśaṅku* to leave quickly so that other Brahmins would not laugh at him. The main reason for *Puṣkarasārin*'s rejection of the courtship was that the *Caṇḍala* caste was not worthy of a Brahmin, as that caste "does not possess the precepts and cannot understand the subtleties of the Vedas" and "the Brahmins do not associate with them" (Ch. 1 of ZKA, p. 402, a23–24.). The traditional Indian texts, such as Rig Veda's "*Puruṣasūkta* (Song of the cosmic being)", have four castes that arise from different parts of the cosmic being's mouth, arms, legs, and feet, while the *Manusmṛti* treatises and other dharma texts prescribe how people of different castes should live with regard to various aspects of social life, such as clothing, food, housing, marriage, birth, and death. Without being condescending, *Triśaṅku* used his knowledge of the Vedas to redefine the origins of caste, arguing that the four castes were simply four brothers born of one mother who were engaged in different occupations and that the other outcastes were the same, with no distinction between higher and lower castes.

The knowledge of the Vedas is available to people today in many ways, but in ancient India, only the three so-called regenerate castes of Brahmins (*Brāhmaṇa*), Kshatriyas (*Kṣatriya*), and Vaishyas (*Vaiśya*) had the opportunity to learn it, and only Brahmins could teach it. The Brahmins held the knowledge, others who had the desire to know were not permitted to learn, and most could not read or write[7]. In particular, the more special knowledge, such as astrological divination and medicine, was read, recited, composed, and applied in practice only by special Brahmin families and was not known to ordinary Indians. After impressing *Puṣkarasārin* with his lectures on Vedic sources, *Triśaṅku* went on to teach a dozen more topics on astrological prophecy, which completely convinced the great Brahmin. The astrological divinations and the astronomical calendar included the names and characteristics of the *nakṣatra*s (lunar mansions); the fraction of days and nights, the length of hours, and the fraction of moments; the unit of length, the weight unit of gold, and the volume unit of grain; the fate of those born on the day of the night; the divination of cities built on the day of *nakṣatra*; the divination of rainfall in the last month of the summer on the day of *nakṣatra*; the divination of lunar eclipses on the day of *nakṣatra*; the desirable and undesirable events on the day of *nakṣatra*; the fraction of days of *nakṣatra*, the length of shadows and the change of hours on the day of *nakṣatra*; and the divination of earthquakes. These divinations are presented one by one in a dialogue between *Puṣkarasārin* and *Triśaṅku*, similar to the format of some intellectual texts. The astrological divinations are based on Vedic astrology, an early stage of Indian astronomy. Some of the divinations in the ZKA text are very similar to those in the *Garagasaṃhitā*, which dates from around the second century AD.

*Puṣkarasārin* was so impressed by *Triśaṅku*'s profound knowledge that he finally gave his daughter in marriage to *Triśaṅku*'s son without a second thought. As a result of this, Ananda and the *Mātaṅga* girl were husband and wife for five hundred lifetimes in the past, living in love and harmony. In this life, Ananda became a disciple of the Buddha, and the *Mātaṅga* girl became a bhikshuni. In a previous life, the Buddha was King *Triśaṅku*;

Ananda was the son of King *Triśaṅku*; and the *Mātaṅga* girl was the daughter of *Puṣkarasārin*, whose mother was the great Brahmin *Puṣkarasārin*. The castes of the Buddha, Ananda, and the mother and the *Mātaṅga* girl were reversed in their previous lives and present lives. The Buddha, as a wise man who knew all the causes of the world, was known as the "World Solver". He knew everything, and after shaving the *Mātaṅga* girl, he told the story of Ananda's past life with the *Mātaṅga* girl to appease the discontent of various groups, including the rest of the monastic community, King *Prasenajit*, and his subjects.

Each of the four conflicting stories mentioned above has great contradictory tension. The *Mātaṅga* woman's desire to break the caste barrier and marry Ananda would have been difficult to achieve in ancient Hindu society, where the caste system was deeply entrenched and daily life was heavily regulated. Because of Ananda's Buddhist identity and his philosophy of the equality of all beings, Ananda's act of asking the *Mātaṅga* girl for a drink of water crossed the barriers of the caste system and won the girl's heart. However, as a monk, Ananda had to follow the many precepts of the Buddhist monastic order: not to speak too much to women; not to live, walk, sit, or lie together; and not to have lustful desires for them. He could not respond to the love of the *Mātaṅga* girl and had to look to the Buddha. The Buddha's solution was to incorporate the *Mātaṅga* girl into the monastic community and make her a *bhikṣuṇī*. However, after becoming a *bhikṣuṇī*, he then had to explain to the disciples, including King *Prasenajit*, why he had accepted a woman from a lower caste into the monastic order. The precepts of the monastic order came into conflict with the resolution of the matter of the *Mātaṅga* girl's pursuit of Ananda. The mother of the *Mātaṅga* girl, because of her daughter's bitter pleading, used a mantra to capture Ananda in the hope of getting him to give in and marry her daughter, turning the wish into reality. The use of the power of language is quite common in Indian culture. Religions such as Hinduism and Buddhism, which also originated in India, have countless stories of the use of mantras and have even formed several sects featuring them. The exclusive knowledge of caste, the Vedas, astrology, etc., was originally held by special Brahmin families and was not often passed on. The Buddhist sutra, the ZKA, has a low-caste king using expertise such as astrology to overcome the difficulties of courtship and marry the daughter of a great Brahmin for his son. Even the long and systematic nature of astrological knowledge gives the impression that this is an early Indian textbook on astrology. The discourse of expertise also becomes a distinctive feature of the ZKA. The four story conflicts mentioned above converge in the ZKA to drive one story climax after another. In fact, any one of the four conflicts is enough to create a good narrative work. The convergence of the above four elements helps the work to shine and spread for over 1700 years.

## 4. Conclusions: Viewing the Buddhist Narration and Logic from the Story of the "Mātaṅga Girl"

The *Śārdūlakarṇāvadāna*, as one of the Sutras of the *Divyāvadāna*, has some similarities to other Buddhist sutras but also some differences. It is a sutra that begins with the words "Thus have I heard" and ends with the words "The bhikṣus rejoiced after hearing the words of the World Honoured One". The narrator is the same as Ananda, but the actual author is the *avadānika* (metaphorist). The narrator in the text is not necessarily the true narrator of the whole story. According to studies by Bangwei Wang and Jingjing Fan, the Indian avadānikas in Buddhist history may have been the true compilers of the *avadāna* stories. Between the first two centuries BCE and the second century CE, there was a widespread trend among Buddhist scholars to make Buddhism more acceptable to a wider audience by increasingly using so-called *avadāna*s in the manner and form of their preaching to produce classics, and these people gradually came to be known by the name of *dārṣṭāntika*s (Wang 2014, p. 78). There are two Sanskrit equivalents of a metaphorist, *dārṣṭāntika* and *avadānika*. *Dārṣṭāntika* may favor the use of exemplary (*dṛṣṭānta*) statements, while *avadānika* tends to use the three-life karma story (*avadāna*) statements. However, the two terms perhaps reflect the succession of historical eras: when the early collections of metaphorical tales were still relatively diverse and mixed, the metaphorist was *dārṣṭāntika*; when they developed

into the standard three-life karma story (*avadāna*), the metaphorist was *avadānika* (Fan 2020, pp. 124–25).

In terms of the narratees of the narratives, the readers of the sutras, in general, are the four members of the monastic community, laymen, and common people who have an interest in Buddhism. *Avadāna* stories arise primarily for the purpose of declaring good and evil. During the unfolding of a narrative, the similarities or differences between the narratee and the narrator, between the narratees and characters, between the narratees and other narrators, and between the narratees and the real reader and the distance between the narratees and the real reader can all be changed. The temporal distance between the narrator and the narratees can change, affecting the tone of the narrative and the development and main idea of the story.

The ZKA has circulated in different linguistic texts, including Sanskrit, Chinese, and Tibetan, and it has been carried out with some local adaptations. But its ability to spread across time and space relies on the uniqueness of its differences from other Buddhist texts. The first point is that the *Śārdūlakarṇāvadāna* covers ancient Indian astronomy and can be regarded as a sort of astronomical textbook, an expertise usually held by a few Brahmin families in India. The second point is that the narrator of the ZKA, Ananda, is also the male protagonist of the story of both his past life and present life. The third point is that the readership of the ZKA may have included divination enthusiasts, astronomical and calendrical researchers, and ordinary people with a practical need for divination.

The fourth uniqueness is that the ZKA brings together four kinds of story conflicts: the conflicts of breaking the caste barrier, of breaking Buddhist precepts, of using a mantra to capture Ananda, and of overcoming the difficulty of seeking marriage with knowledge. The above four kinds of story conflicts are intertwined, resulting in a climactic reading experience. The breaking of the caste barrier reflects the Buddhist concept of the equality of all beings. However, a monk followed the precepts of the Buddhist monastic order and could not respond to the love of the *Mātaṅga* girl. The girl's mother tried to solve the problem by using a mantra to capture Ananda to make their marriage happen. The solution to the problem was to incorporate the *Mātaṅga* girl into the monastic order as a *bhikṣuṇī*. Then, the Buddha had to explain in past lives why it was possible for a low-caste woman to enter the monastery. In the story of a previous life in the ZKA, a low-caste king used expertise such as astrology to overcome the difficulties of courtship and marry the daughter of a great Brahmin for his son. Ananda and the *Mātaṅga* girl were married for 500 lifetimes before they became entangled in this life. The four interlocked links drive the storyline forward and conclude satisfactorily by responding to the opening conflict.

Whether it is a literary story with clear contradictions, an astrological text with a significant readership, or a mantra that is always mystical, it can be broadly assumed that the main purpose of the ZKA in combining these elements was to make Buddhism more widely available and accessible to a wider audience. According to the research, the real source of the modern Chinese word "mó dēng 摩登", as a transliteration of the English word "modern", would be the Buddhist story "*Śārdūlakarṇāvadāna*", which was popular in areas along the Silk Road more than 1700 years ago. Perhaps, while people's group memories fade away as individual lifespans come to an end, living creations are still able to transcend language, region, and time, displaying a different luster and gaining constant life in a new time and space.

**Funding:** This research was funded by Major Project of the National Social Science Foundation of China: Indian Art and Literary Theories in Classical Sanskrit Literatures: Translation and Studies on Fundamental Works (中国国家社会科学基金重大项目"印度古典梵语文艺学重要文献翻译与研究"), and the grant number is 18ZDA286.

**Institutional Review Board Statement:** Not applicable.

**Informed Consent Statement:** Informed consent was obtained from all subjects involved in the study.

**Data Availability Statement:** Not applicable.

**Acknowledgments:** I am grateful to Imre Galambos of the University of Cambridge for inviting me to engage in an academic exchange that resulted in a prototype of the thesis. I am grateful to Jingjing Fan of PKU, for sharing her monograph and to Research Librarian Lina Wang of the National Library of China and Can Li of BFSU for taking time out of their busy schedules to correct this thesis. Sincere thanks to Chenfeng Zhu for checking my English draft. Boundless thanks to all the reviewers and editors who offered pages of suggestions and teachings. All errors should be attributed to myself and are reflective of the further improvements that can be made in my study of this subject.

**Conflicts of Interest:** The author declares no conflict of interest.

## Abbreviations

| | |
|---|---|
| ZKA | *Śārdūlakarṇāvadāna* in Sanskrit. |
| Ch 1 of the ZKA | 《大正新脩大藏經》T21, No. 1300, 摩登伽經（題为竺律炎、支谦譯）. |
| Ch 2 of the ZKA | 《大正新脩大藏經》T21, No. 1301, 舍頭諫太子二十八宿經（竺法护譯）. |
| ZKA in *Divyāvadāna* | *dūlakarṇāvadāna* in The *Divyāvadāna: A Collection of Early Buddhist Legends*, ed. Edward Byles Cowell and Robert Alexander Neil, Cambridge 1886: The University Press, pp. 611–55. |
| ZKA from Nepal | *The Śārdūlakarṇāvadāna*, ed. by Sujitkumar Mukhopadhyaya, Santiniketan 1954: Viśvabharati. |

## Notes

[1] The storyline is narrated according to *Śārdūlakarṇāvadāna*, Ch 1 of the ZKA, i.e., Modengjia jing (摩登伽經), and its English translation The Mātaṅga Sutra (Giebel 2015), as well as some other parallel texts (Cowell and Neil 1886; Mitra 1882; Mukhopadhyaya 1954; Vaidya 1959; Hiraoka 2007; Tatelman 2005).

[2] Cf. (Fan 2020, pp. 71–72), the idea of Matsumura comes from Hisashi Matsumura, *Four Avadānas from the Gilgit Manuscripts*. The Australian National University, 1980. Diss., pp. XXXIX–XL.

[3] Cf. (Jiang 2007, p. 206), "Shudra and the Vaishya girl, with the Kshatriya girl and the Brahmin girl, would give birth to a mixed race of Ayoghas, Kshatris and *Caṇḍala*, the lowest of men. 蔣忠新譯《摩奴法論》第十章第12條" 首陀羅與吠舍姑娘、與刹帝利姑娘和婆羅門姑娘所生的是雜種性阿約格弗、刹德利和人中最低賤者旃陀羅。".

[4] Cf. Ch 1 of ZKA, p. 403, c23–24, (Giebel 2015). "Once there was also a person who was traveling along the road when his carriage broke down, whereupon he repaired it, and so he was called *mātaṅga* 時複有人，于路遊行，其車破壞，因便修治，名摩登伽。" Cf.Ch 2 of ZKA, p.414a9–11, "Once there was also a woman who was traveling along the road of the wild, broke others' carriage who become auspicious or unauspicious then, and she was called *mātaṅga* 有一婦人，行在異路曠野屏處，破壞車轂，眾人吉凶，是故世間得凶呪種。".

[5] The mantra is from the Sanskrit text *Śārdūlakarṇāvadāna*, not from Ch 1 of Modengjia jing 摩登伽經. I took a reference from Joel Tatelman's draft of the English translation of the ZKA, Heavenly Exploits 33, and checked (Giebel 2015) on the English translation of Ch 1.

[6] The second mantra here is from the Sanskrit text *Śārdūlakarṇāvadāna*, not from Ch 1. I took a reference of Joel Tatelman's draft of the English translation of the ZKA, Heavenly Exploits 33, and checked (Giebel 2015).

[7] In the 2011 Indian census, the average literacy rate in India was 74%, 82.14% for males and 65.46% for females; at the end of British colonial rule in 1947, the average literacy rate in India was 12%; how can one presume that the literacy rate in ancient India was no higher than 12%?

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
