# Peer review of "How Did Bhikṣuṇī Meet Indian Astrology? Viewing the Buddhist Narration and Logic from the Story of the Mātaṅga Girl"

_religions, doi:10.3390/rel14050657_

Round 1

Reviewer 1 Report

This paper makes a thorough survey of the famous Buddhist text Śārdūlakarṇāvadāna, focusing on the story of the Mātaṅga girl, especially from the perspective of narratology.

After talking about the general characteristics of the Buddhist literary genre avadāna, the paper analyses the special features of the Śārdūlakarṇāvadāna, including its story-line and characters, paying close attention to the four conflicts in the main plot. I believe the author has satisfactorily addressed most of the readers’ concerns in a well-structured manner.

Three small suggestions. Firstly, maybe the author could provide an English translation of the two mantras in Lines 309-312 and Lines 318-320, so that the reader might better understand those mantras. Secondly, the English translations of 撰集百缘经 in Line 120 and Line 139 are different. It would be better to choose one as the standard translation. Thirdly, the pinyin of 摩登伽 should also be the same throughout the paper. But now, in Line 103 it is Modejia, while in Line 198 it is Modenjia. I think the standard form might be Modengjia.

Besides, I found a typo. In Line 169, Divāvadāna should be Divyāvadāna.

In my opinion, this paper deals with a very interesting topic and is quite charming. I like it and recommend the manuscript for publication after minor revision.

Author Response

Dear Reviewer,

A million thanks for your recommendation and suggestions. I've revised my draft as follows:

Point1:  maybe the author could provide an English translation of the two mantras in Lines 309-312 and Lines 318-320, so that the reader might better understand those mantras.

Response 1: Firstly,I provide an English translation of the two mantras in Lines 309-312 and Lines 318-320, so that the reader might better understand those mantras.

Point2: the English translations of 撰集百缘经 in Line 120 and Line 139 are different. It would be better to choose one as the standard translation.

Response2: Secondly, I choose one standard translation of 撰集百缘经 in Line 120 and Line 139 .

point3: Thirdly, the pinyin of 摩登伽 should also be the same throughout the paper. But now, in Line 103 it is Modejia, while in Line 198 it is Modenjia. I think the standard form might be Modengjia.

Response3: Thirdly, all  pinyins of 摩登伽  were checked and changed to Modengjia. 

Point4:  I found a typo. In Line 169, Divāvadāna should be Divyāvadāna.

Response4: Besides, the  typo you found in Line 169, Divāvadāna was corrected as Divyāvadāna.

It's glad that you point out the mistakes in English, Chinese, and Sanskrit. 

Sincerely author

Reviewer 2 Report

"though corresponding to the English word modern,"

What does this mean? The modern meaning as a transliteration of "modern" isn't relevant to a discussion of the Buddhist text.

triśaṅku needs to be capitalized (Triśaṅku) as it is proper name.

"to marry his daughter, Benjamin"

Where is Benjamin in the text? Is this a typo?

舍头諌太子二十八宿经 (竺法护)

Use 繁體字 and not 簡體字.

"Modejia jing (translated by Zhulvyan and Zhiqian) 摩登伽51 经(题为竺律炎+支谦译), Ch2 of ZKA, i.e. Shetoujian taizi ershibaxiu jing (translated by 52 Dharmarakṣa Zhufahu) 舍头諌太子二十八宿经 (竺法护), Foshuo modengnv jing (trans-53 lated by An Shigao) 佛说摩邓女经(题为安世高译), Foshuo modengnv jiexing zhong 54 liushi jing (anonymous) 佛说摩登女解形中六事经 (失译); Nepalese Sanskrit in the 17th 55 century Śārdūlakarṇāvadāna; Tibetan translation in the ninth century sTag rNa'i rTogs pa 56 brJod pa, etc."

Check pinyin, italicize all pinyin titles. 

The attributed translator of the 摩登伽經 is disputed. See the following:

Kotyk, Jeffrey. 2017. “Iranian Elements in Late-Tang Buddhist Astrology.” Asia Major 30, no. 1: 25–58. See pages 28-29.

Use "ü" not "v" in pinyin: nü 女, not nv 女.

"Caṇḍala. The Chandala caste"

The Sanskrit transcription needs to be used consistently.

"Brahmin, Kshatriya, Vaishya and Shudra"

Better to use Sanskrit diacritics consistently throughout.

"This is because the ancient Hindu Dharma..."

Which source? Cite a canonical authority. How about Manusmṛiti?

"The Manusmṛti states that "the dwelling place of Caṇḍalas and 211 Shivbhagas must be outside the village, they must be treated as mendicants, and their 212 possessions must be dogs and donkeys."

Whose translation is this? What part of the text? Cite the Sanskrit ideally.

"Buddha used his magical powers to know that Ananda was confined and used a spell to 271 break the Brahmin woman's spell and Ananda was able to return to the monastery."

Doesn't this happen with a mantra?

"bhikkhuni"

This is Pali. Why use this term over the Sanskirt? The author uses Sanskrit throughout the paper. bhikṣuṇī is the Sanskrit term (nun). bhikṣu (monk).

"Generally speaking, a bhikkhu takes 253 precepts, some of which are forbidden to 279 "change one's mind through lust"."

The different vinayas have different precept numbers. There's no general number in reality, unless you mean Theravada, but the present paper is not working with a Theravadin text.

"Her mantras was “amale vimale kuṅkume sumane/yena 309 baddhāsi vidyut/icchayā devo varṣati vidyotati garjati/vismayaṃ mahārājasya sa-310 mabhivardhayituṃ devebhyo manuṣyebhyo gandharvebhyaḥ śikhigrahā devā viśikh-311 igrahā devā ānandasyāgamanāya saṃgamanāya kramaṇāya grāṇāya juhomi svāhā//”."

Translate or explain this?

What is the source of this? Which manuscript? Page numbers?

"In Indian culture, people believe in the power of the body, speech and mind."

this is more a Buddhist idea related to karma. A very broad statement about "Indian culture" -- is this applicable 2000 years ago as it is today?

The author should consider citing at least some of these studies (especially those of Zenba Makoto). The author doesn't cite much in the way of secondary sources. There's actually a lot that needs to be referenced.

Aoyama Tōru 青山亨. 1982. “Śārdūlakarṇāvadāna no kenkyū.” Indogaku bukkyōgaku kenkyū印度学仏教学研究  60 (30–2): 152–153.

Cowell, Edward B,  Robert Alexander Neil, eds. 1886. The Divyâvadâna: a Collection of Early Buddhist Legends. Cambridge:  Cambridge University Press. https://archive.org/details/pts_divyvadnacol_3720–0688

Hiraoka Satoshi 平岡聡. 2007. Budda no nazo toku sanze no monogatari: Diviya Avadāna zenyakuブッダが謎解く三世の物語『ディヴィヤ・アヴァダーナ』全訳 . Tokyo:  Daizō shuppan. 2 volumes

----. 2002. Setsuwa no kōkogaku: Indo bukkyō setsuwa ni himerareta shisō說話の考古学:インド仏教說話に秘められた思想 . Tokyo:  Daizō shuppan.

Hiraoka, Satoshi. 2011. “The Divyāvadāna in English.”  In Indo-Iranian Journal. vol. 54 (3)231–270.

Mitra, Rajendralala. 1882. The Sanskrit Buddhist Literature of Nepal. Calcutta:  The Asiatic Society of Bengal. https://archive.org/details/sanskritbuddhist00asiauoft

Miyazaki, Tensho. 2015. “The Śārdūlakarṇāvadāna from Central Asia.”  In Karashima, Seishi,  Margarita I. Vorobyova-Desyatovskaya, eds. Buddhist Manuscripts from Central Asia The St. Petersburg Sanskrit Fragments (StPSF). Tokyo:  The Institute of Oriental Manuscripts of the Russian Academy of Sciences and The International Research Institute for Advanced Buddhology Soka University. vol. 1 1–84. http://iriab.soka.ac.jp/orc/Publications/StPSF/index_StPSF.html

Mukhopadhyaya, Sujitkumar, ed. 1954. Śārdūlakarṇāvadāna. Santiniketan:  Viśvabharati.

Pingree, David. 1963. “Astronomy and Astrology in India and Iran.”  In ISIS. vol. 54 (2)229–246.

Sharma, Arvind. 1978. “The Puruṣasūkta: Its Relation to the Caste System.” Journal of the Economic and Social History of the Orient 21 (3): 294–303.

Shinjō Shinzō 新城新藏. 1989. Tōyō tenmongakushi kenkyū東洋天文學史研究 . Kyoto:  Rinsen shoten. Reprint of 1928 work

Vaidya, P. L., ed. 1959. Divyāvadāna. Darbhanga:  Mithila Institute of Post-Graduate Studies and Research in Sanskrit Learning. http://www.dsbcproject.org/node/7185

Yabuuchi Kiyoshi 薮内清. 1990. Chūgoku no tenmon rekihō中国の天文暦法 . Tokyo:  Heibonsha.

Zenba Makoto 善波周. 1956. “Butten no tenmon rekihō ni tsuite佛典の天文暦法について .” Indogaku bukkyōgaku kenkyū印度學佛教學研究  7 : 18–27.

Zenba Makoto 善波周. 1952. “Matōga gyō no tenmonrekisū ni tsuite摩登伽經の天文曆數について .”  In Tōyōgaku ronsō: Konishi, Takahata, Maeda san kyōju shōju kinen東洋學論叢:小西高畠前田三教授頌壽記念 . Kyoto:  Heirakuji shoten. 171–214.

Author Response

Dear Reviewer,   sincere thanks for your efforts of pointing out my mistakes and showing a better way of academic writing. I'll remember your teaching and improve my writing from this paper. 

Point 1 "though corresponding to the English word modern,"

What does this mean? The modern meaning as a transliteration of "modern" isn't relevant to a discussion of the Buddhist text.

Response1: transliteration is used in new sentence as followed, The modern Chinese word 'modeng(摩登)', a transliteration of English word “modern”, is indeed borrowed from the ancient Chinese translation of the sutras with its cultural connotations of Indian women(Zhang2007, p.31). 

Point2: triśaṅku needs to be capitalized (Triśaṅku) as it is proper name.

Response2: All triśaṅku were capitalized as Triśaṅku. 

Point3: "to marry his daughter, Benjamin"

Where is Benjamin in the text? Is this a typo?

Response3: It's indeed a typo which was corrected as " to marry his daughter, Prakṛti".

point 4: 舍头諌太子二十八宿经 (竺法护)

Use 繁體字 and not 簡體字.

Response4: All Chinese characters were corrected to Classic Chinese 繁體字. 

Point 5: "Modejia jing (translated by Zhulvyan and Zhiqian) 摩登伽51 经(题为竺律炎+支谦译), Ch2 of ZKA, i.e. Shetoujian taizi ershibaxiu jing (translated by 52 Dharmarakṣa Zhufahu) 舍头諌太子二十八宿经 (竺法护), Foshuo modengnv jing (trans-53 lated by An Shigao) 佛说摩邓女经(题为安世高译), Foshuo modengnv jiexing zhong 54 liushi jing (anonymous) 佛说摩登女解形中六事经 (失译); Nepalese Sanskrit in the 17th 55 century Śārdūlakarṇāvadāna; Tibetan translation in the ninth century sTag rNa'i rTogs pa 56 brJod pa, etc."

Check pinyin, italicize all pinyin titles. 

Response 5: Checked and corrected pinyin, italicize all pinyin titles. 

Point6: The attributed translator of the 摩登伽經 is disputed. See the following:

Kotyk, Jeffrey. 2017. “Iranian Elements in Late-Tang Buddhist Astrology.” Asia Major 30, no. 1: 25–58. See pages 28-29.

Responses6:  Thanks for the reference, I use "supposed" here, as following:  Modengjia jing (supposed to be translated by Zhulvyan and Zhiqian) 摩登伽经(题为竺律炎+支谦)

Point7: Use "ü" not "v" in pinyin: nü 女, not nv 女.

Response7: corrected in the paper

Point8: "Caṇḍala. The Chandala caste"

The Sanskrit transcription needs to be used consistently.

"Brahmin, Kshatriya, Vaishya and Shudra"

Better to use Sanskrit diacritics consistently throughout.

Response8: I've corrected the Sanskrit transcription as much as possible. 

Point9: "This is because the ancient Hindu Dharma..."

Which source? Cite a canonical authority. How about Manusmṛiti?

Responses9: corrected as "because the ancient Hindu Dharma literature like Manusmṛti "

Point 10. "The Manusmṛti states that "the dwelling place of Caṇḍalas and 211 Shivbhagas must be outside the village, they must be treated as mendicants, and their 212 possessions must be dogs and donkeys."

Whose translation is this? What part of the text? Cite the Sanskrit ideally.

Response10: The translation belongs to Jiang Zhongxin, a Chinese Sanskrit scholar.

Point11: "Buddha used his magical powers to know that Ananda was confined and used a spell to 271 break the Brahmin woman's spell and Ananda was able to return to the monastery."

Doesn't this happen with a mantra?

Response11: I've deleted this sentence. 

Point12: "bhikkhuni"

This is Pali. Why use this term over the Sanskirt? The author uses Sanskrit throughout the paper. bhikṣuṇī is the Sanskrit term (nun). bhikṣu (monk).

Response12: correted all, including the title. 

Point13: "Generally speaking, a bhikkhu takes 253 precepts, some of which are forbidden to 279 "change one's mind through lust"."

The different vinayas have different precept numbers. There's no general number in reality, unless you mean Theravada, but the present paper is not working with a Theravadin text.

Response13: I've corrected this sentence. 

Point14: "Her mantras was “amale vimale kuṅkume sumane/yena 309 baddhāsi vidyut/icchayā devo varṣati vidyotati garjati/vismayaṃ mahārājasya sa-310 mabhivardhayituṃ devebhyo manuṣyebhyo gandharvebhyaḥ śikhigrahā devā viśikh-311 igrahā devā ānandasyāgamanāya saṃgamanāya kramaṇāya grāṇāya juhomi svāhā//”."

Translate or explain this?

Response 14: I've added translations of mantras.

Point15: What is the source of this? Which manuscript? Page numbers?

"In Indian culture, people believe in the power of the body, speech and mind."

this is more a Buddhist idea related to karma. A very broad statement about "Indian culture" -- is this applicable 2000 years ago as it is today?

Response15: From ancient texts of India, people believe in the power of the body, speech and mind. In Hindu mythology and literature, episodes in which vows and mantras become reality and cause great hurt to mundane people and even celestial gods are commonly used

Point 16: The author should consider citing at least some of these studies (especially those of Zenba Makoto). The author doesn't cite much in the way of secondary sources. There's actually a lot that needs to be referenced. 

Response16:

Before I chose references only on ZKA stories not on ZKA astrology. Later I will read again your references  you listed. 

Reviewer 3 Report

I have made comments in the margins and highlighted infelicities of language in the attached PDF

Author Response

Dear Reviewer,

sincere thanks for your suggestions on Pdf. I found the end of abstract is really confusing, so I delete the last sentence. Italicized all Sanskrit words which are too many in this paper, I think. 

From author

Reviewer 4 Report

There are some suggestions for modification.

1) Line 48, “he converted to……”, should be “she converted to……”

2) Line 74, the title of section 1.1 is “The Opening and the Conclusion”, however form line 115 to line 141 in this section, the concept of Avadāna is discussed, which is not very relevant to “The Opening” or the “Conclusion”.

3) Line 91-93, “‘Thus have I heard (如是我闻)’ appears in a later translation, used by translator Yi Jing(义净) from the Tang dynasty and translator Shi Hu(施护) from the Song() dynasty, and gradually becoming a standard in the translation of Buddhist texts.” There are two problems with this statement. 1)Thus have I heard (如是我闻)’ actually was used by earlier translator like Buddhayasas佛陀耶舍and Zhu Fonian竺佛念 in The Longer Agamasutra (長阿含經) in 412-13 CE, and Fa Xian (法顯) in The Mahaparinirvanasutra (大般涅槃經) in 416-418 CE, etc.. 2) The meaning of “gradually becoming” is not very clear. Semantically, it seems to indicate that it “gradually” became a standard after Shi Hu of Song dynasty. While the translation of Buddhist sutras to Chinese was stopped in the early Song Dynasty of Shi Hu’s time.

4) Line 347, the section number “3.4” should be “2.4”.

5) Line 432-498, Part III looks like a conclusion of the whole paper. I suggest simply making it a conclusion, so that readers can more clearly understand the conclusions of this paper. Then the text of part III should be rewritten to be more concise and clear. Some details of the argument, likes the emerging of avadānikas (line 438-450), can be adjusted to the previous text.

Author Response

Dear Reviewer,

sincere thanks for your points which help me to improve this paper. 

Point1: Line 48, “he converted to……”, should be “she converted to……”

Response1: thanks for she. 

Point2: Line 74, the title of section 1.1 is “The Opening and the Conclusion”, however form line 115 to line 141 in this section, the concept of Avadāna is discussed, which is not very relevant to “The Opening” or the “Conclusion”.

Response2: Opening here I mean  " Thus Have I heard".  I did some changes here like, 1.1. The Opening and Ending Phrases.

Point3: Line 91-93, “‘Thus have I heard (如是我闻)’ appears in a later translation, used by translator Yi Jing(义净) from the Tang dynasty and translator Shi Hu(施护) from the Song(宋) dynasty, and gradually becoming a standard in the translation of Buddhist texts.” There are two problems with this statement. 1)‘Thus have I heard (如是我闻)’ actually was used by earlier translator like Buddhayasas佛陀耶舍and Zhu Fonian竺佛念 in The Longer Agamasutra (長阿含經) in 412-13 CE, and Fa Xian (法顯) in The Mahaparinirvanasutra (大般涅槃經) in 416-418 CE, etc..

2) The meaning of “gradually becoming” is not very clear. Semantically, it seems to indicate that it “gradually” became a standard after Shi Hu of Song dynasty. While the translation of Buddhist sutras to Chinese was stopped in the early Song Dynasty of Shi Hu’s time.

Response3: I've reorganized the sentences here. 

Poin:4 Line 347, the section number “3.4” should be “2.4”.

Response4: 3.4 corrected to 2.4. Thanks.

point5: Line 432-498, Part III looks like a conclusion of the whole paper. I suggest simply making it a conclusion, so that readers can more clearly understand the conclusions of this paper. Then the text of part III should be rewritten to be more concise and clear. Some details of the argument, likes the emerging of avadānikas (line 438-450), can be adjusted to the previous text.

Response5: The avadānikas part was adjusted to the conclusion part as it's not proper to be placed there. I wish that I could place here while I do some revisions on conclusion.